# Clinical Relevance of *Helicobacter pylori* Infection

**DOI:** 10.3390/jcm10163473

**Published:** 2021-08-06

**Authors:** Irena Mladenova

**Affiliations:** Medical Faculty, Department of Hygiene, Epidemiology, Microbiology, Parasitology and Infectious Diseases, Trakia University, 6000 Stara Zagora, Bulgaria; imladenova@yahoo.com; Tel.: +359-897-324472

**Keywords:** *Helicobacter pylori*, chronic gastritis, peptic ulcer disease, gastric cancer, MALT-lymphoma, therapy, vaccines

## Abstract

*Helicobacter pylori (H. pylori)* is a Gram-negative helical, microaerophilic bacterium which colonizes the antrum and body of the stomach, surviving in its harsh environment through mechanisms of acid resistance and colonization factors. It infects approximately 50% of the world population. Although the prevalence of this infection varies from country to country, as well as between different ethnic, social or age groups, it is estimated that about 50% of the human population only carries this microorganism. While *H. pylori* has been found to play a major etiological and pathogenic role in chronic gastritis, peptic ulcer disease and gastric cancer, its importance for many types of extra-gastric disease needs to be further investigated. The choice of tests to diagnose *H. pylori* infection, defined as invasive or non-invasive, depends on the clinical indication as to whether to perform upper gastrointestinal endoscopy. Focusing on bacterial eradication, the treatment should be decided locally based on the use of antibiotics and documented antibiotic resistance. The author provides an overview of the current state of knowledge about the clinical aspects of *H. pylori* infection, especially its diagnostic and therapeutic management.

## 1. Introduction

The identification of *Helicobacter pylori (H. pylori)* by researchers Warren and Marshall in 1982 revolutionized the concept of gastric inhospitality and the consideration of peptic ulcer as a noninfectious disease. This microorganism is a Gram negative helical, microaerophilic bacteria which colonizes the antrum and body of the stomach, surviving in its harsh environment through mechanisms of acid resistance and colonization factors [1]. Through the enzyme urease, the microorganism creates a cloud of acid neutralizing substances around itself, offering protection from the acid [2]. Since *H. pylori* infects more than 50% of the world’s population, it represents one of the most common infections in humans, usually acquired in the preschool period, with a risk of acquisition declining after 5 years of age and influenced by poorer living conditions during childhood [3].

Although the prevalence of this infection is different in all countries and also between ethnic, social, or age groups, much higher rates of *H. pylori* infection have been reported in developing countries than in developed countries. Nevertheless, it is important to highlight that only a minority of infected people develops health issues and life-threatening diseases [3].

In this paper, the author provides an overview of the current state of knowledge about the clinical aspects of *H. pylori* infection, especially its diagnostic and therapeutic management.

## 2. Clinical Impact of *H. pylori* Infection

### 2.1. Gastroduodenal Diseases

#### 2.1.1. Gastritis and Peptic Ulcer

*H. pylori* is a major etiological and pathogenic factor for chronic gastritis, peptic ulcer (PUD) and gastric cancer [3]. In a recent review, a total of 55 randomized controlled trials and long-term treatments of peptic ulcer disease were included. The authors concluded that one or two-week eradication treatment of H. pylori infection is a very effective therapy for *H. pylori*-positive patients with duodenal ulcer compared to ulcer healing drugs alone or without therapy [4]. The Kyoto Global Consensus Meeting has been convened to reach a global consensus on the classification of chronic gastritis and duodenitis, to differentiate *H. pylori*-induced dyspepsia from functional dyspepsia, to adequately diagnose gastritis, and to help clinicians decide when, whom and how to treat. An anonymous electronic consensus-building system has been adopted using the Delphi method. In order to better organize the definition of gastritis and duodenitis based on the etiology, the experts recommended a new classification of these conditions. In particular, an innovative diagnostic algorithm of *H. pylori*-associated dyspepsia has been proposed [5]. 

Classifications of chronic gastritis have been offered over time. Currently, pathologists mainly follow the recommendations of the revised Sydney System. OLGA (Operative Link for Gastritis Assessment) and OLGIM (Operative Link for Gastritis Intestinal Metaplasia) assessments reported by pathologists are used by clinicians to differentiate patients with chronic gastritis for special monitoring [6]. Those who have chronic gastritis without *H. pylori* infection should be considered to have a disease caused by a previous *H. pylori* infection [7].

To understand the diseases associated with *H. pylori.* it is important to study the virulence factors. Lately, factors have been reported for the colonization HopQ, SabA, BabA, OipA, and necessary factors necessary which are a sign of a pathogenicity island, such as vacuolating cytotoxin A (VacA), cytotoxin-associated gene antigen (CagA), and the outer membrane vesicles [8,9]. Several studies have evaluated the prevalence of *H. pylori* CagA and VacA genotypes, assessing the relationship with the type of damage caused by gastroduodenal mucosa. In patients with PUD, the genotype’s vacA s1 cagA-positive strain has a close relationship. The vacAs1 subtype has been detected in all patients. Hence, VacA s1 is an important marker of virulence and patients harboring these strains are more likely to develop ulcers. VacA s1 could serve as the only best marker for the virulence of *H. pylori* [10].

#### 2.1.2. Dyspepsia

The prevalence of functional dyspepsia in the general population is 10–20%. It is a functional disorder of the gastrointestinal tract. Typical dyspeptic symptoms are nausea, epigastric pain, and a feeling of fullness. Proton pump inhibitors (PPIs) and *H. pylori* eradication are the most commonly used treatment. Impaired quality of life in patients with functional dyspepsia suggests the need for definitive diagnosis, followed by symptomatic treatment and preventive management of relapse [11,12].

Functional dyspepsia is often associated with *H. pylori* infection. A patient who has an endoscopy without a pathology is defined as having functional dyspepsia and *H. pylori* therapy must be offered in case of infection [13]. Patients with dyspepsia and *H. pylori* infection may have clinical features that distinguish them from those with functional dyspepsia but without infection. In some studies, the authors evaluated the existence of clinical differences between uninfected individuals with functional dyspepsia and those with *H. pylori* infection and dyspepsia.

The prevalence of *H. pylori* in 578 dyspeptic patients without significant lesions detected by endoscopy was 58% (divided into two groups, i.e., positive or negative for *H. pylori*). Cases of dyspepsia and *H. pylori* infection have been associated with obesity, blood pressure, diabetes, and metabolic syndrome, but finally the paired analysis negated all the differences. Thus, dyspeptic patients positive for *H. pylori* have the same clinical features as uninfected ones [14].

Many randomized trials have investigated the effects of *H. pylori* eradication in patients with functional dyspepsia. The pooled estimates were measured by the fixed or random effect model from 25 randomized controlled trials in 5555 patients with functional dyspepsia. The pooled risk ratio (RR) was 1.23 (95% confidence interval (CI): 1.12–1.36). *H. pylori* eradication induced improvement in symptoms during more than 1 year follow-up (RR = 1.24, *p* ˂ 0.0001) but not with follow-up time less than 1 year (RR = 1.26, *p* = 0.27). Seven studies did not show a benefit from *H. pylori* eradication on quality of life. In six studies, eradication therapy reduced the incidence of PUD. Ten studies reported that patients who received eradication therapy had higher probability of experiencing histological regression of chronic gastritis than untreated ones [15]. 

Thus, the eradication of *H. pylori* in patients with functional dyspepsia has to be individual evaluated [16].

#### 2.1.3. Gastric Cancer

*H. pylori* infection is considered to be the biggest risk factor for stomach cancer, a disease that takes hundreds of thousands of lives a year. This bacterium was classified as a group I carcinogen in 1994 by the World Health Organization (WHO) [3]. 

Approximately 75% of the global burden of gastric cancer and 5.5% of malignancies worldwide are due to inflammation and injury caused by *H. pylori*. It has been shown that *H. pylori* eradication therapy reduced the incidence of gastric cancer in high-risk areas. In Japan the treatment against *H. pylori* infection is effective to prevent stomach cancer (Lin et al.) [17]. However, the extent of benefit of the same approach in people living in areas with different prevalence of gastric cancer remains unclear. A meta-analysis included randomized controlled trials to assay the effects of eradication therapy on the risk of developing gastric cancer. People with *H. pylori* eradication had a lower incidence of gastric cancer than those without eradication therapy. The baseline incidence of gastric cancer altered the benefit of *H. pylori* eradication. Eradication provided significant benefit to both asymptomatic infected patients and to ones followed by endoscopic resection for gastric cancer. Therefore, there was a link between the elimination of *H. pylori* infection and the reduced incidence of the stomach cancer. The benefits of eradication varied depending on the incidence of gastric cancer but were associated to the baseline risk [18].

As stomach cancer is one of the leading cause of cancer-related deaths worldwide, the approach to preventing this malignancy is a very important public health issue. Gastric cancer is associated with an inflammatory tumor with multistage and multifactorial carcinogenesis. The process includes a series of steps with development of metaplastic epithelium, dysplasia and gastric cancer. *H. pylori* infection is critical for the development of the disease and studies have consistently shown that bacterial eradication reduces inflammation of the gastric mucosa, stopping the progression of the atrophy, metaplasia and dysplasia and lowering the risk of PUD onset and carcinogenesis development. The screening and eradication of *H. pylori* have recently begun only in high-risk populations. Elimination of gastric cancer requires information for implementing effective *H. pylori* screening and treatment programs, taking into account other health priorities in each particular population [19].

The pathogenesis caused by *H. pylori* is mainly attributed to its virulence factors, including urease, flagella, VacA, and CagA. The last two factors, VacA and CagA, play a key role. Infection with vacA-positive strains of *H. pylori* can lead in the stomach mucosa to vacuolation and apoptosis, while infection with CagA-positive strains can result in severe gastritis and gastric cancer. Studies focused on gastric carcinogenesis divide risk factors into categories such as host responses, genotypes, strain variation, and environmental factors. By assessing the interactions between these factors, we can understand the risk and progression of the disease in people with persistent colonization [20].

The CagA protein is an oncoprotein that can induce malignancies in mammals. On delivery, CagA disrupts multiple host pathways by acting as a scaffold. CagA-induced gastric carcinogenesis progresses through a shock-triggering mechanism in which the prooncogenic actions of CagA are followed by a series of genetic or epigenetic changes composed of cancer-prone cells during long-term infection with CagA-positive *H. pylori* [21]. Identifying high-risk individuals is important for monitoring and preventing stomach cancer. The presence of first-degree relatives diagnosed with gastric cancer is a strong risk factor for gastric cancer. Тhe pathogenic mechanisms are unclear. There is an increased risk of developing stomach cancer among patients having two or more affected first-degree relatives with a family history of *H. pylori* infection. Eradication of *H. pylori* is the most important strategy to prevent stomach cancer in first-degree relatives of patients with stomach cancer. Early eradication of *H. pylori* can prevent progression to intestinal metaplasia and reduce the possibility of developing gastric carcinogenesis in these individuals [22].

It is recommended to adopt classification systems for stratification of gastric cancer risk and modern endoscopy to improve the image for the diagnosis of gastritis. Eradication therapy against *H. pylori* prior to the development of preno-plastic changes has been recommended to minimize the risk of severe complications from this infection [5]. These changes could be an important factor in identifying high risk individuals for gastric cancer [23]. Patients undergoing endoscopic treatment for gastric cancer are at high risk of developing metachronous gastric cancer. Patients with precancerous lesions who do not reverse after treatment with *H. pylori* are at the ‘point of no return’ and may be at high risk of developing stomach cancer. An earlier eradication of *H. pylori* should prevent the development of gastric cancer before the onset of precancerous lesions [24,25].

The authors of one study recommended, for previous cases of atrophic gastritis caused by *H. pylori*, endoscopic monitoring every 3 years for high risk patients, including those with endoscopic severe atrophy or intestinal metaplasia [18]. Japan introduced a strategy for *H. pylori* eradication in 2013 to reduce the number of new cases of gastric cancer and deaths caused by this malignancy and, respectively, the medical costs. It was estimated that the number of deaths from stomach cancer could be reduced to 30,000 per year by 2020, but the annual number of deaths in 2017 remained above 45,000. The effect of the strategy may not appear until 2023. The risk of gastric cancer is likely to increase in some populations due to the widespread use of PPIs and dysbiosis in the stomach mucosa. In one study a combined therapy with PPIs and aspirin has been proposed after the eradication of *H. pylori* [26]. To reduce the incidence of stomach cancer health promotion has to be included, including adequate physical activity, low alcohol intake, dietary nutrition, and quitting smoking [27,28].

In prospective studies conducted by immunoblot assay (compared to those using ELISA-based methods), the worldwide attributed fraction for *H. pylori* in non-cardia gastric cancer has increased to 89.0% (6.2% of all cancers). In this way, the role of *H. pylori* as a main cause of gastric cancer has been enhanced [29]. The incidence and case fatality rate from stomach cancer are higher in developing countries than in developed countries. The prognosis of gastric cancer is much poorer because of its diagnostic delay. Approximately 2% of *H. pylori*-positive individuals develop gastric cancer [30].

#### 2.1.4. Gastric Mucosa-Associated Lymphoid Tissue (MALT) Lymphoma

The low-grade B-cell lymphoma, called the mucosal associated lymphoid tissue (MALT) lymphoma, is formed in the stomach in response to antigenic stimulation, primarily associated with *H. pylori* infection. There is a strong relationship between *H. pylori* infection and low-grade MALT lymphoma. Furthermore, *H. pylori* eradication in patients with low-grade MALT lymphoma leads to tumor regression [31]. *H. pylori* strains of gastric MALT lymphoma appear to be less virulent than those associated with PUD or gastric carcinoma. A specific antigenic profile of Lewis has been identified in these strains and may represent an alternative mechanism to avoid the immune response of the macro-organism, thus allowing a continuous antigenic stimulation of the lymphocytes in the tissue [32].

In France, in a population-based study, the clinical characteristics and survival of patients with MALT lymphoma were analyzed. Among 460 confirmed patients only 44 showed early transformation into diffuse large B-cell lymphoma and were thought to have initially missed high-grade lymphoma. *H. pylori* was detected in 57% of the cases. Eradication was obtained in 76% of patients and complete remission occurred in 70%. The overall 5-year survival was 79% [33]. The diagnosis of MALT lymphoma has to be posed by endoscopic biopsy confirmed by an experienced pathologist. There are many variable endoscopic pictures such as erosion, lesion, atrophy, and ulcer, so many biopsies are needed to make an accurate diagnosis. Eradication therapy is the basis of treatment in all patients, at all stages of the disease. If remission does not occur after eradication therapy, radiation therapy or chemotherapy should be given. In the case of advanced disease, immunotherapy with an anti-CD20 monoclonal antibody may be used [34].

## 3. Extra-Gastroduodenal Diseases

Many studies have shown that *H. pylori* infection can influence the onset of several extra-gastroduodenal diseases. The role of this bacteria in idiopathic thrombocytopenic purpura and iron deficiency anemia is currently well documented. Emerging data suggest that it may also contribute to insulin resistance, diabetes mellitus, non-alcoholic liver disease, and metabolic syndrome. In addition, it may increase the risk of coronary artery disease and neurodegenerative disease [35,36].

Several meta-analyses have been conducted on the potential association between cardiovascular disease and *H. pylori* infection. Meta-analytical approaches with fixed and random effects have been performed. The findings suggest a possible link between *H. pylori* infection and the risk of myocardial infarction [37,38]. Another field of investigation regards special populations, such as patients with chronic kidney disease, who present gastric mucosal injuries and dyspepsia more often than the general population. These diseases have a multifactorial pathogenesis and *H. pylori* infection could play a limited role in their development [39].

## 4. Diagnostic Methods

### 4.1. Initial Diagnosis

When endoscopy is required, the current diagnostic invasive approaches are biopsy and histology, immunohistochemistry, urease detection, culture assay, and polymerase chain reaction (PCR). The implementation of sequencing technologies is subject to the recommended guidelines for the management of *H. pylori* infection. The determination of the gold standard among all methods remains controversial, especially for epidemiological studies. Because of the declining sensitivity of invasive tests, non-invasive tests, including serology, stool antigen test and urea breath test, have been largely used for detecting *H. pylori.* Urea breath test and stool antigen test, among the non-invasive tests, are the best methods to detect the active infection. The sensitivity of serology tests is high but the specificity is relatively low. The guidelines show that no test can be considered the gold standard for diagnosing H. pylori and there are advantages and disadvantages of all methods [40,41,42,43].

### 4.2. Confirmation of Eradication

Confirmation of *H. pylori* eradication is always recommended. Stool antigen test and urea breath test can be used to confirm eradication when endoscopy is not required and have to be accomplished at least 4 weeks after the end of the therapy [43,44,45]. Since it is known that PPIs exert transient negative effects on *H. pylori* viability, morphology, and urease test, cessation of these drugs at least 14 days before testing for eradication could help avoid false-negative results (Maastricht V) [46].

## 5. Treatment of *H. pylori* Infection

### 5.1. First-Line Treatment

Considering the choosing of the regimen for *H. pylori* eradication, previous exposure to antibiotics should be accounted. The triple clarithromycin-based therapy must be confined to patients without prior exposure to macrolides living in areas with a low resistance to clarithromycin. At present, bismuth quadruple therapy or concomitant non-bismuth quadruple therapy (PPI, amoxicillin, clarithromycin and nitroimidazole) should be the preferred regimen. This has been shown to be most effective in overcoming antibiotic resistance. After the failure of the first-line therapy, the rescue regimen should avoid antibiotics that have been used before. If the patient has received first-line treatment containing clarithromycin, the preferred treatments are bismuth salts schemes or levofloxacin. Treatment with levofloxacin, which is known as a second-line therapy after treatment with clarithromycin, should also be recommended after failure of quadriceps containing bismuth salts [47,48].

Recent key topics include studies to evaluate the effectiveness of bismuth quadruple therapies. Now there is strong evidence that it is the best first-line therapy in most countries. In fact, antibiotic resistance has been comprehensively studied and a drastic increase in resistance to clarithromycin and levofloxacin has been noted [49]. The utility of vonoprazan (a competing potassium acid blocker) instead of PPI therapy has also been considered, especially in resistant and difficult-to-treat groups. However, presently this drug is used only in Japan. The diversity of the intestinal microbiota is altered soon after the eradication of *H. pylori* with triple therapy and has been restored after 2 months [50].

In 2017, the WHO identified as a high priority for research the resistant *H. pylori* strains to clarithromycin. Resistance to metronidazole and fluoroquinolones has also increased worldwide [51]. The international consensuses for *H. pylori* eradication recommended quadruple therapy with bismuth or non-bismuth for 2 weeks as a first-line treatment in areas with a high resistance to clarithromycin and/or metronidazole [52]. These schemes provide good levels of eradication. A new approach is needed to reduce antimicrobials and to protect against resistance in case of dual therapy. A good option is vonoprazan and amoxicillin. This could be a breakthrough in the era of increasing antibiotic resistance. This scheme can provide an acceptable degree of eradication, reduces antimicrobial resistance because of the use of single antibiotic and includes an inhibitory effect of vonoprazan on the acid secretion of the stomach. Assessed in a first period only in Japan recently, its efficacy has also been reported in Australia. In a single-center study, conducted in the period January 2017–September 2019, treatment with vonoprazan-containing antibiotic therapy was capable of achieving 100% efficacy in patients treated for the first time and even 91% efficacy in patients with previous eradication failure (Gunaratne et al. [53]).

If the treatment of *H. pylori* fails, new approaches are needed. A meta-analysis aimed to study the role of symbiotics in eradication therapy. A random effects model has been applied to the pooling analysis due to the heterogeneity of the studies. The results have shown that the symbiotic may improve eradication with RR: 1.28. Frequent adverse effects from the antibiotic therapy have been significantly diminished by adding a symbiotic to standard antibiotic treatment. The meta-analysis has suggested that symbiotics might improve the eradication rate of *H. pylori* infection, and reduce the adverse effects [54].

### 5.2. Second-Line Treatment

The failure of the first-line therapy for *H. pylori* infection requires a second-line therapy which is challenged due to potential microbiological resistance to the antibiotics included initially [55]. There is no “golden” standard in rescue eradication therapy after failure of first-line treatment. The advice of the Maastricht V/Florence Consensus Report is in favor of quadruple bismuth therapy or triple/quadruple fluoroquinolone–amoxicillin therapy as а second-line therapy [46]. Meta-analyses proved that the eradication results of quadruple bismuth therapy and levofloxacin–amoxicillin therapy are almost equal, while the first has more adverse effects than the second. The rate of eradication of triple and quadruple levofloxacin-based therapies is suboptimal.

In case of fluoroquinolone resistance, the triple or quadruple levofloxacin–amoxicillin therapy has a lower efficacy of eradication. A 10-day therapy consisting of PPI, bismuth, tetracycline and levofloxacin was recently developed, which achieved a significantly higher degree of eradication compared to triple therapy with PPI–levofloxacin–amoxicillin (98% vs. 69%) in patients after failure of standard therapies [55,56].

As a second-line treatment, the tetracycline–levofloxacin, bismuth-based or levofloxacin–amoxicillin quadruple therapies could be administered for *H. pylori* eradication. Recent data suggest that 10-days tetracycline–levofloxacin therapy is an effective scheme and a candidate for rescue treatment after failure of eradication by all first-line schemes for *H. pylori* infection. A document comparing the recommendations in the guidelines of expert groups in Europe, Canada and the United States has been published. The guidelines recommend bismuth quadruple therapy for first-line treatment, replacing triple clarithromycin-based therapy. In case of unsuccessful treatment, because of the resistance to antibiotics or another drugs, they must be avoided in eradication therapy. Second-line therapies have to be quadruple bismuth therapy; triple levofloxacin therapy, a triple scheme based on rifabutin or amoxicillin in high doses plus PPI due to the suspicion of resistance, can be used for subsequent treatment [57,58].

### 5.3. Third-Line (and Further) Treatment

After two failed therapies, susceptibility-guided treatments have been administered as a third-line strategy. This could be a rescue treatment. Nevertheless, evidence in favor of this therapy is insufficient and the cure rate is moderate [58]. The efficacy of the third-line therapies for *H. pylori* is suboptimal, even after a bacterial culture. Resistance to many antibiotics is the main factor for treatment failure. The effectiveness and safety of 2-weeks eradication using high doses of amoxicillin, metronidazole and esomeprazole in patients with two previous failures of therapy has been assessed. Triple therapy with esomeprazole 40 mg twice daily, amoxicillin 1 g three times daily and metronidazole 500 mg three times daily for 14 days has been implemented as third-line therapy after first therapy, including clarithromycin, and a second-line treatment, including quinolone [59].

The microbiological resistance against antibiotics is increasing worldwide and the rate of failure of *H. pylori* first and second eradication lines is increasing. The role of cultural assay in testing of antibiotic susceptibility is very important to avoid the use of ineffective therapy. There are many causes of eradication failure, including poor compliance of patient with the eradication scheme, smoking, or factors related to treatment such as doses and length of therapy. Treatment could be modified specifically for the respective patient. Individuals at high risk of developing gastric cancer may receive definitive benefits after third or fourth line therapy [60].

Since there is the possibility that *H. pylori* eradication fails, in this case the indication to continue PPI treatment should be considered in patients at high risk for PUD complications, such as at increasing age and those who need a long term treatment with gastrolesive drugs such NSAID and for patients taking anticoagulation (Kanno and Moayyedi) [61].

### 5.4. Adding an Adjuvant Treatment

Probiotics have inhibitory effect on *H. pylori* and have been used as adjunctive therapy in *H. pylori* eradication. Probiotics have improved eradication rate of *H. pylori* and side effects of antibiotic treatment. Treatment outcomes are conflicting due to species, doses, and length of administration. Additional studies on the safety of adjuvant probiotics in eradication therapy of *H. pylori* are needed [62,63,64].

In one randomized placebo-controlled study the efficacy of probiotics as an adjuvant to eradication therapy of *H. pylori* has been evaluated. A total of 159 patients receiving sequential eradication therapy against *H. pylori* were included. The degree of elimination, patient compliance, and side effects of eradication therapy were recorded in each treatment group. Adjuvant application of a probiotic in 14 days sequential therapy for *H. pylori* has been associated with a higher rate of *H. pylori* eradication, lower rates of discontinuation associated with diarrhea, less frequent side effects and higher adherence to treatment [65].

## 6. Relapse and Reinfection

Recurrence can occur either through relapse or through reinfection. To determine relapse or reinfection, and to match the treatment and follow-up of patients to the nature of relapses, it is mandatory to study genotype [66]. Compared to reinfection, the relapse time window is usually shorter, followed by a recurrence of *H. pylori*-related diseases. Reinfection after an effective eradication therapy is very rare [67]. Several factors are responsible for *H. pylori* reinfection, such as the presence of *H. pylori* positive family members, poor living conditions, and health status. The factors for *H. pylori* relapse need further study [68].

## 7. Vaccine

Unfortunately, there is not an effective vaccine against *H. pylori*. This could be the best weapon against *H. pylori* and prevention of gastric cancer, respectively. A radical change in therapeutic strategies is needed to guide the final decisions about the management of *H. pylori.* The unique nature of *H. pylori* creates obstructions to the development of an immunogenic vaccine. In developing countries, the most reasonable and logical approach would be to recommend a preventive vaccine against *H. pylori* among children as a mandatory national program to reduce the likelihood of early acquisition of infection. Attempts to produce a prophylactic vaccine will be postponed to the future [69]. The modulating of the host immune responses by *H. pylori* results in increasing regulatory T cell proliferation. It is possible to generate protective immune responses by immunization with various *H. pylori* antigens or their epitopes, in combination with an adjuvant, so far only shown in mouse models. New non-toxic adjuvants have recently been developed, consisting of modified bacterial enterotoxins or nanoparticles, which can not only increase the efficacy of the vaccine but also help vaccines to be applied in clinical practice [70].

Knowledge of the immune mechanisms during *H. pylori* infection due to the host’s complex response to the pathogen and the factors that allow bacterial persistence, such as *H. pylori* genetic diversity, are needed for an effective vaccine [71]. It would be a strong tool to prevent gastric cancer. Despite the high prevalence of the infection worldwide and evidence that vaccination can prevent children from acquiring *H. pylori* infection, the development of such a vaccine is not a current priority for major pharmaceutical companies. More investment is needed to step up research into the *H. pylori* vaccine [72,73]. This has to involve immunizing mice with classical and recombinant *H. pylori* antigens in order to develop а vaccine against *H. pylori.* Efficacy is very difficult to prove, usually involving many clinical trials. Promising results have been reported by Ming et al. in 2015 [74].

## 8. Conclusions

In clinical practice, since a decision must be made with therapeutic intent when the bacteria *H. pylori* is found, treatment should be based on topical antibiotic use and documented data on antibiotic resistance. In countries with a high rate of clarithromycin resistance (>15%), in the first line therapy, either ‘concomitant’ or bismuth-based quadruple therapies are recommended as empirical treatment if antimicrobial susceptibility testing is not possible. In the second-line therapy, as with empirical treatment, if antimicrobial susceptibility testing is not possible, quadruple therapy which was not used as a first-line treatment is recommended. Triple therapy combining PPI with amoxicillin and levofloxacin is also possible. Drugs such as rifabutin and furazolidone should be reserved for further steps.

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
