# Peer review of "Clinical Relevance of Helicobacter pylori Infection"

_jcm, 2021, doi:10.3390/jcm10163473_

Round 1
Reviewer 1 Report
Dear author,
This review article presented updating information about Helicobacter pylori focusing on clinical diseases. The article was written in plain English and possibly informative to clinicians. However, some of the discussion may be somewhat out of regular clinical implications based on smaller reports. So you need to consider reconsidering the text in line with stronger evidence-based citations as below or need to add more information.
1. For the "Gastritis and peptic ulcer" section
Eradication therapy for peptic ulcer disease in Helicobacter pylori-positive people.
Cochrane Database Syst Rev. 2016. PMID: 27092708 Free PMC article. Review. 2. For the "Dyspepsia" section ACG and CAG Clinical Guideline: Management of Dyspepsia. Am J Gastroenterol. 2017 Jul;112(7):988-1013. DOI: 10.1038/ajg.2017.154. Epub 2017 Jun 20.PMID: 28631728 Review. 3. For the "Gastric cancer" section Need appropriate citation after "... reduced the incidence of gastric cancer in high-risk areas". 4. For the "Confirmation of eradication" You may need to add comments about false-negative risk under PPI intake. 5. For the last paragraph on the " Treatment of H. pylori infection" section There is a certain probability that the eradication of H. pylori will fail. Therefore, I think it would be more useful for readers to indicate in which cases, such as the failure of eradication, PPIs must be continued. For example, you may refer to the following review article. Who Needs Gastroprotection in 2020? Curr Treat Options Gastroenterol. 2020 Nov 11:1-17. DOI: 10.1007/s11938-020-00316-9. Online ahead of print.PMID: 33199955 Free PMC article. Review.
Author Response
I thank the Reviewer for the comments and suggestions which give me the opportunity to improve this manuscript. Please find enclosed the revised version of this article, with changes tracked according to requirements of the Journal and a letter with the point by point reply.
Point-by-point reply
- For the "Gastritis and peptic ulcer" section: I have added the reference- Ford, A.C.; Gurusamy, K. S.; Delaney, B.; Forman, D.; Moayyedi, P. Eradication therapy for peptic ulcer disease in Helicobacter pylori-positive people. Cochrane Database Syst Rev. 2016, 4:CD003840, highlighted in yellow.
- For the "Dyspepsia" section: I have added the reference- Moayyedi, P.; Lacy, B.E.; Andrews, C.N.; Enns, R.A.; Howden, C.W.; Vakil, N. Management of Dyspepsia. Am J Gastroenterol. 2017, 112, 988-1013, highlighted in yellow.
- For the "Gastric cancer" section after "... reduced the incidence of gastric cancer in high-risk areas": I have added the reference- Lin, Y.; Kawai, S.; Sasakabe, T. et al. Effects of Helicobacter pylori eradication on gastric cancer incidence in the Japanese population: a systematic evidence review. Japanese Journal of Clinical Oncology, 2021, 51, 1158–1170, highlighted in yellow.
- For the "Confirmation of eradication": “Since it is known that PPIs exert transient negative effects on H. pylori viability, morphology, and urease test, cessation of these drugs at least 14 days before testing for eradication could help avoiding false-negative results (Maastricht V)”.
- For the last paragraph on the "Treatment of H. pylori infection": I have added the reference- Kanno, T.; Moayyedi, P. Who Needs Gastroprotection in 2020? Curr Treat Options Gastroenterol. 2020, 11,1-17, highlighted in yellow.
Reviewer 2 Report
The review concerns an important issue which is gastrointestinal infection caused by Helicobacter pylori. Most of the text is generally available information. The Author of the paper should emphasize and focus on the works recently published so that the manuscript can-pretend to be original and novel.Author Response
I thank the Reviewer for the comments and suggestions which give me the opportunity to improve this manuscript. Please find enclosed the revised version of this article, with changes tracked according to requirements of the Journal and a letter with the point by point reply.
Point-by-point reply
Since substantial changes have been reported in the field of eradication therapy, I added recent Australian data regarding the advantage in using vonoprazan in antibiotic regimens. This is a relevant data because it reports that this drug could change positively the results not only in Japan. Furthermore, since for clinicians it is crucial to know how manage patients with H.pylori eradication failure I reported recent literature suggestions on this cohort as well as in those who need of long-term treatment with drugs inducing gastrolesive effects.
I have added the reference: Gunaratne, A.W.; Hamblin, H.; Clancy, A, et al. Combinations of antibiotics and vonoprazan for the treatment of Helicobacter pylori infections—Exploratory study. Helicobacter. 2021;:e12830. https://doi. org/10.1111/hel.12830, highlighted in yellow, and some more references from 2020/2021.
Round 2
Reviewer 1 Report
Dear Author,
This review articled added some informative clinical pieces of evidence, Now this one could be considered as accepted. The author should change citation numbers in order before publication.
Author Response
Dear Reviewer,
I have changed the citation numbers in the order in which they appear in the text.
Thank you very much for your help and suggestions!
Reviewer 2 Report
The Author addressed and responded to all questions raised during the first evaluations of the manuscript
Author Response
Dear Reviewer,
Thank you very much for your help and suggestions!
This manuscript is a resubmission of an earlier submission. The following is a list of the peer review reports and author responses from that submission.